# Morphological Detection Of Helicobector Pyloric Organisms On Gastric Mucosa Using Deep Learning Of The Artificial Intelligence

**Shuanlong Che**
KingMed Diagnostics, Guangzhou, China.
`shuanlong2008@sina.com`

**Chao Li**
Semptian Co., Ltd., Shenzhen, China.
`landlee.work@gmail.com`

**Pifu Luo**
KingMed Diagnostics, Guangzhou, China.
`0pedac001@qq.com`

**Longsen Chen**
Semptian Co., Ltd., Shenzhen, China.
`chenlongsen@semptian.com`

## Abstract

Since discovered, Helicobacter pylori (H. pylori) is acknoledged as one of the major causes of gastric ulcer, duodenal ulcer, gastritis, and gastric cancer. Detecting the infection of H. pylori in human body is of great significance for the treatment of various gastrointestinal diseases caused by Helicobacter pylori. This paper provides a new method of dectecting H. pylori based on digital pathology and deep learning. Through the study of a large number of whole slide images (WSIs), the detection of H. pylori on WSIs from gastric biopsy is achieved. The experimental results in this paper show that this method can achieve good detection performance and has certain promotion and practical value.

## 1 INTRODUCTION

Helicobacter Pylori is a type of unipolar, multi-flagellate, obtuse-ended, spirally curved bacterium with negative Gram-negative, 2.5-4.0 $\mu$m in length and 0.5-1.0 $\mu$m in width. H. pylori is the only bacterium known to survive in stomach acid, and it is active near the pyloric region of the stomach. H. pylori infection can cause a variety of gastrointestinal diseases, and greatly increase the chance of suffering from gastric cancer. The gold standard for clinical diagnosis of H. pylori infection is to perform pathological examination. The gastric mucosal tissue taken from biopsy will be stained and made into slides for doctor observation under the high magnification microscopy. Hematoxylin-eosin (HE), methylene blue, and silver stain are the staining methods for gastric mucosa. The most commonly used and most suitable for routine examination are HE staining. Under HE staining conditions, H. pylori usually appears pink or light red.

With the continuous advancement of computer science, deep learning technology has made remarkable breakthroughs in many fields such as image, video and speech recognition in recent years. In the field of image recognition, the Convolutional Neural Network (CNN) is very prominent and can even surpass the normal human performance in many specific scenarios. In the field of digital pathology, the use of deep learning technology for auxiliary diagnosis is a very promising direction of development.

This paper proposes a method based on deep learning to achieve H. pylori detection in HE stained whole slide images (WSI), and the results are marked on WSIs for review by pathologists. The WSIs mentioned in this paper are all scanned on 40-magnification. The CNN model used is CIFAR-10[1], the software platform is tensorflow, and the main hardware is two TITAN XPs.

1st Conference on Medical Imaging with Deep Learning (MIDL 2018), Amsterdam, The Netherlands.

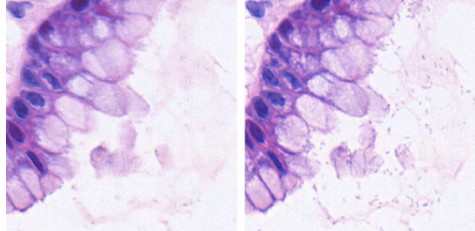

Figure 1: H. pylori sharpness contrast under different scanning methods

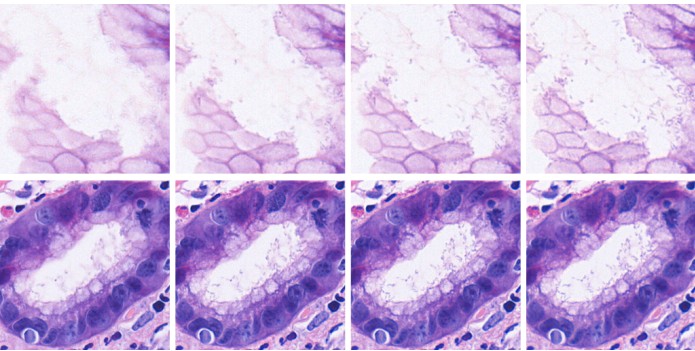

Figure 2: Multi-layer scan images in the same field of view after the sharpness evaluation

## 2 METHODS

### 2.1 Image generation

The gastric mucosal tissue slides used in this study were from a domestic medical-related institution. Strict desensitization was performed on the slides, and all were HE stained. The digital scanner used in this paper was configured for 40x scanning magnification, with pixel space of 0.243 $\mu$m and a depth of field of 0.25 $\mu$m.

Training a CNN model requires a large number of labeled training samples taken from the WSIs of the gastric mucosal tissue. Usually, the length and width of these images can be as high as 100,000 or even more, but the sample patches extracted from it can be much smaller.

During the process of scanning the slide, we found that there is a problem with the WSIs obtained by using the ordinary scanning method. A large number of H. pylori are too vague to be visually recognized. The reason for this phenomenon is that a H. pylori is very small in size and has different positions on the tissue slides. The depth of field of the scanner objective lens is limited, and it is difficult to clearly photograph all H. pylori distributed at different heights. We use multi-layer scanning to solve this problem. Multi-layer scanning is performed at a fixed interval on the Z-axis of the scanner so that the depth of field can cover the entire height of the slide. Therefore, there are always 1 2 layers that can be scanned clearly. In this paper, five-layer scans were used, with a 0.5 $\mu$m interval between layers. After each slide was scanned, five layers of WSIs were obtained. The comparison between the single-layer scanned image and the clearest layer out of the multilayer scanned images is as figure 1:

Under the same visual field of 5 layers of image sha evaluation, sorting effect as shown in figure 2:

### 2.2 Data generation

The pixel space of the scanner used in this paper is 0.243um, so the H. pylori length is only a dozen pixels at most, and the width is about three pixels. Considering the valid proportion of the images occupied by the H. pylori targets, the input image size of the CIFAR-10 neural network is taken into account. It is reasonable to define the size as 32×32; in order to reduce the distortion of the

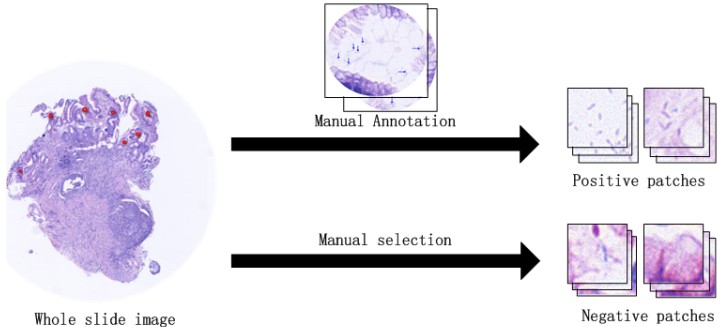

Figure 3: Training set generation method

original picture information, the sample pictures in this paper are saved in the lossless compressed PNG format.

The sample data of the training set is divided into positive and negative pictures. We call these pictures "patches". A positive patch is a picture containing one H. pylori or more, and a negative one is a picture without any H. pylori. The rules for generating training patches set come from the doctor's annotation and review. Firstly, a pathologist accurately annotate the H. pylori coordinate information on the WSIs of the training set slides. Then, with each marked coordinate as the center, a fixed-size patch is extracted at each layer of the multi-layer WSI. The evaluation algorithm[2,3] picks out the clearest image in the same coordinate and adds to positive training set. The negative patches are generated from regions without any H. pylori. To ensure the sample's richness and comprehensiveness, the superficial, deep and marginal regions of the tissue are sampled at the same time, and the proportion is balanced as far as possible. This is shown in figure 3.

The method of manual annotation by pathologist is highly accurate, but it takes a long time, resulting in slow generation of positive training patches. In this paper, the method of generation of training patches employs a combination of manual and mechanical methods. First of all, pathologists manually annotate the earliest part of the data in training set, then we train a small model based on the CIFAR-10 network. Although the inference accuracy of this small model is not high enough, it can select out the patches with high probabilities of positive from a large number of patches. All these suspicious positive patches are then handed over to pathologists for review. After the false positive patches are removed, the remaining patches are added to the positive train set. In this way, the inference accuracy of the small model is continuously improving, and the probability of the selected sample being reviewed is also increasing. Therefore, the sample generation speed will be faster and faster. Using this method to produce positive patches, the rate is more than 8 to 10 times higher than manual annotation.

The patches of the test set is taken from the multi-layer scanned WSIs of the slides from test set. Firstly, the area of the gastric mucosal tissue regions in a WSI is identified by the back-and-front separation algorithm[4]. Then two types of regions, the gland lumens and the neighboring regions surrounding tissue, are identified based on the previously identified tissue regions. Finally, within the two types of regions, all coordinates of available patches are calculated in the size of $32 \times 32$ pixels, and the clearest picture at each coordinate is extracted from the multi-layer WSIs using a sharpness evaluation algorithm. All the generated patches are added to the test set. It should be noted that in order to prevent the occurrence of missed detection due to the fact that a H. pylori is split into two adjacent patches, the size of grid stride should be taken as 16, which increases the number of samples to approximately 4 times. However, the possibility of missed detection is greatly reduced. About 100,000-500,000 patches are extracted per WSI, and the number of patches is positively correlated with the size of the tissue. The method of regions identification will be described in detail in section 2.5.

## 2.3 Data augment

In the process of training CNN, sample is preprocessed to achieve data enhancement. The data augment methods used in this paper include random cropping, horizontal flipping, random rotation,

color dithering, and so on. Because of the small size of a H. pylori, patches containing H. pylori can not be zoomed in or out during training. Otherwise, the characteristics of H. pylori itself will be seriously distorted. In order to ensure that the positive patches always include H. pylori after being randomly cropped, an annotated coordinate should be the center of a positive patch, and the size of the training patches is $48\times48$, the size of the cropped patches is $32\times32$, and no scaling is performed. Horizontal flipping performs a random horizontal mirroring operation on the training patches, and the minimum granularity for random rotation is 90 degrees. The range of luminance jitter is 10%.

## 2.4 CNN training

This paper built a two-category model based on CIFAR-10 network, and software is based on Tensorflow implementation. CIFAR-10 is a 5-layer CNN network that contains two convolutional layers, two full-connection layers, and a softmax linear layer. Each layer uses relu as activation function, and the output layer uses softmax for classification. The learning parameters of the entire network are nearly 200,000 and it belongs to a small network, which is suitable for rapid classification of small size pictures.

The network parameters are updated in gradient descent method, the batch size is 256, the initial learning rate is 0.1, the attenuation coefficient is 0.1, and it decays once every 13,000 steps. In order to improve the generalization ability of the network, this paper modifies the loss function used in network training and increases the L2 regularization of the weight parameters of the convolutional layer, the fully connected layer, and the softmax linear layer. The experiment results on the verification data set prove that this method can improve the recognition accuracy while reducing the steps required for network training. The modified loss function is shown as below:

$$
\begin{aligned}
Loss_{total} = L_{softmax} &+ 0.004 \times L2_{conv1} + 0.004 \times L2_{conv2} + 0.004 \times L2_{fc1} \\
&+ 0.004 \times L2_{fc2} + 0.004 \times L2_{softmax\_linear}
\end{aligned}
\tag{1}
$$

In order to speed up the training, we firstly convert the patches of training to examples and then packet them into multiple files in TFRecord format. We create a MIMO queue during training, create multiple threads to parse the examples from TFRecord files and add them to the MIMO queue. We also create a number of threads to take examples from the MIMO queue, perform data augement and transfer them to CNN network for training.

## 2.5 Machine detection

### 2.5.1 H. pylori activity regions detection

H. pylori activity regions are lumens and the neighboring regions surrounding tissue. Identifying these effective activity regions for machine identification will, on the one hand, help improve the accuracy of detection, and on the other hand will help reduce invalid test patches and shorten detection time. According to experimental statistics, the detection time can be reduced by more than 10 times. In this paper, the detection of valid activity regions is divided into three stages. The first one is tissue region detection, the second is lumen detection, and the third is detection of neighboring regions surrounding tissue. We merge the second and third regions as H. pylori activity regions. The flow of tissue regions detection is shown as figure 4.We extract the thumbnail from one layer of a WSI, convert the color space of thumbnail from RGB to Yuv, calculate the threshold of u channel using Otsu algorithm, and binarize the thumbnail according to the calculated threshold to obtain a mask image of the tissue regions. We call this mask image as foreground mask. The last picture in figure 4 is a mask image of tissue regions.

The lumen regions detection flow is shown as figure 5. In the first stage we have obtained the foreground mask of tissue regions. The foreground contains the regions where the nucleus, cytoplasm, and other physiological tissues are located. The components of background are more complicated, including the blank area of the slide, the cavity of lumens, and the gap between sparse tissue. The vast majority of sparse tissue gaps are relatively small in area and can be removed by morphological methods. After removing the blank areas, mask for lumens are obtained.

The detection of the neighboring regions is shown in figure 6. In the first stage we have obtained the foreground mask of tissue regions, and the foreground in mask represents the tissue regions. All the closed background areas inside the foreground mask are filled to obtain a solid tissue mask. Due to



Figure 4: The tissue regions detection flow

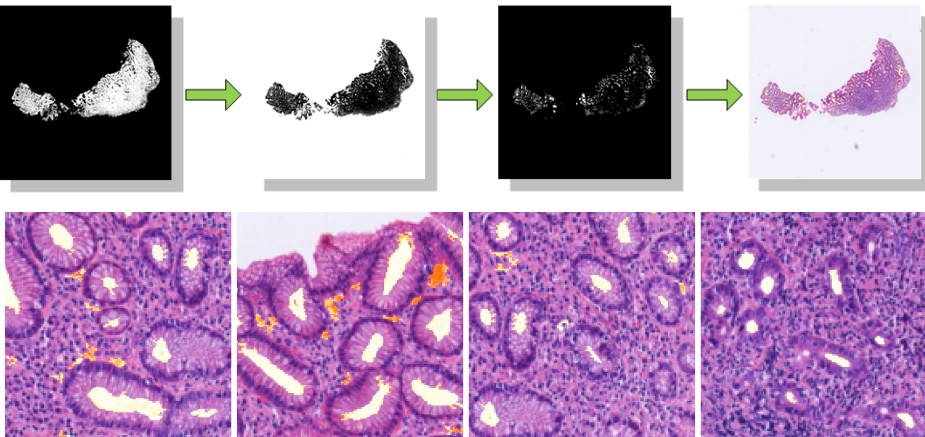

Figure 5: The Lumen regions detection flow and results. The four pictures in the first row show the flow of lumen regions detection, and the four pictures in the second row show the results. Each detected lumen is marked with yellow color

the existence of sparse tissue areas in some slides, that is, there are more gaps between the tissues, the mask will generate multiple adjacent areas that are not connected. These disconnected areas are in deeper smooth muscle tissue, not H. pylori activity regions, so we remove them by morphological connectivity; other small isolated tissues are not H. pylori activity regions and are also removed; The mask with the above content cleared is binary dilated to obtain the mask of the neighboring regions surrounding a slide.

### 2.5.2 H. pylori detection

H. pylori detection identifies only the H. pylori activity regions described in Section 2.5.1. Firstly, the H. pylori activity regions of each test WSI are cut in a grid. The image size is 32×32 pixels, and the cutting stride is 16 pixels. Overlap cutting is used to prevent the H. pylori on the edge of the cutting grid from being cut in half and located on two adjacent patches. Because the half of H. pylori in

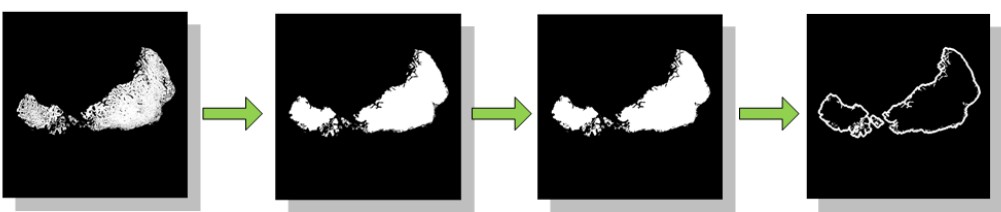

Figure 6: The neighboring regions detection flow

Table 1: Test set data statistics

| data set name | slide number | patch number | ground truth for patches |
|---|---|---|---|
| test set | 9 | 3490012 | P=34514, N=3455498 |

Table 2: Test results

| Ground Truth / Detection Result | Positive | Negative | |
|---|---|---|---|
| Positive | 33514 (TP) | 334 (FP) | 33848 |
| Negative | 1000 (FN) | 3455164 (TN) | 3456164 |
| | 34514 | 3455498 | |

each patch does not include the complete H. pylori features and is prone to be missed detection. The resulting patches are sent to CIFAR-10 network previously trained in Section 2.4 for reasoning. The results on the validation set prove that setting the positive threshold to 0.7 is a reasonable choice. This paper uses two NVIDIA TITAN X network inferences to process images at a rate of about 5000 patches/GPU/sec.

# 3 EXPERIMENTS AND RESULTS

## 3.1 Evaluation metrics

This paper uses 9 test slides to evaluate system performance. Since the average patch number cut from a WSI can be up to 600,000, the workload for producing ground truth labels for patches of the test slides is huge, so far only 9 WSIs are selected for testing. The test WSIs were cut into patches according to Section 3.5, and a negative or positive label was assigned to each patch after manual review. The evaluation metrics for the test are sensitivity and specificity. The specific formulas are as follows:

$$Sensitivity = \frac{TP}{TP + FN} \qquad (2)$$

$$Specificity = \frac{TN}{TN + FP} \qquad (3)$$

## 3.2 Results

The detailed data of the test set is shown in table 1:

The test results are shown in table 2:

The sensitivity and specificity are calculated as below:

$$Sensitivity = \frac{33514}{34514} = 97.1\%$$

$$Specificity = \frac{3455164}{3455498} = 99.99\%$$

The reasoning result of the patches are displayed on the WSIs as figure 7, and each red bound-box in the figure represents a positive patch after cutting.

# 4 CONCLUSIONS

This paper provides an efficient H. pylori detection method based on deep learning and digital pathology. This method can solve the problem of H. pylori vagueness in the ordinary scanning mode, and effectively locate the H. pylori activity regions to improve the speed of detection of WSIs. The method of this paper achieves a high sensitivity and specificity on the test set, with reference research and practical value.

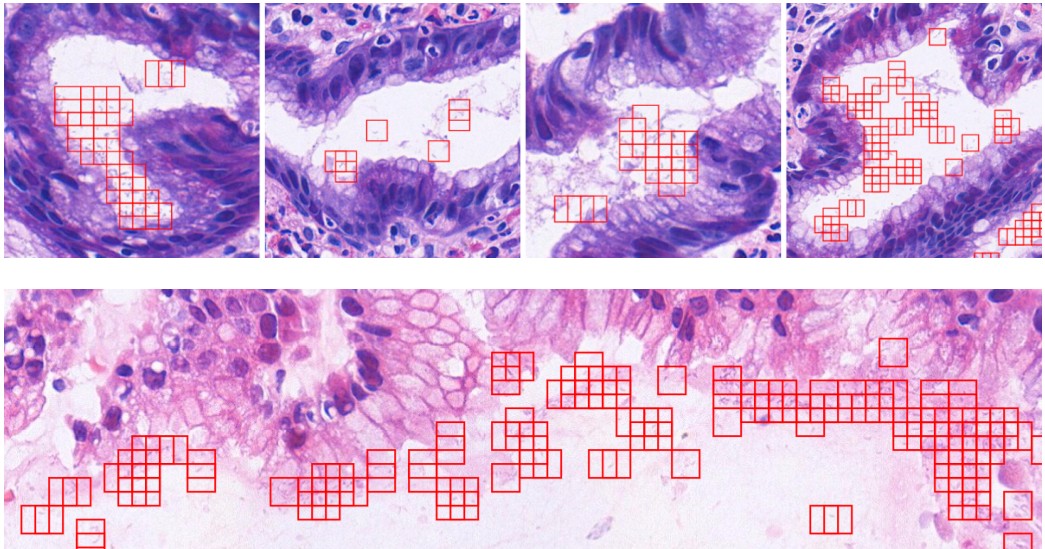

Figure 7: Positive patches displayed on the WSIs. We used overlapping cuts to prevent H. pylori from losing features due to cutting, so the same H. pylori will be detected simultaneously by multiple adjacent patches.

## 5  DISCUSSION

This paper uses the sharpness evaluation method to obtain the most clear field of vision in a multi-layered WSI. This is a method to solve the sharpness of H. pylori imaging; another method is to fuse the multi-layer scanned images into a single-layer image. The image merges the clearest field of view in each layer, so it can be processed directly as a single-layer image, eliminating the need for sharpness evaluation methods. Based on the detection of H. pylori patches, the counting and density statistics of H. pylori in the whole digital image can bring more convenience to pathologists, which is the direction of this paper.

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
