# OpenReview forum: "Morphological Detection Of Helicobector Pyloric Organisms On Gastric Mucosa Using Deep Learning Of The Artificial Intelligence"
_MIDL.amsterdam/2018/Conference — Submitted to MIDL 2018_

### Review · AnonReviewer2 · 2018-04-26
**Automated Detection Of Helicobector Pyloric Organisms On Gastric Mucosa**

**Rating:** 2
**Confidence:** 3

**Review:**

This paper introduces a methodology for the automated detection of Helicobector Pyloric (H. pylori) organisms on gastric mucosa using standard deep learning models.  The idea is to take a whole slide image (WSI) scanned at 40x magnification, sample image patches of size 32x32 pixels, which are then classified into positive (i.e., containing one or more H. pylori organisms) or negative.  This image sampling process is done with a few pre-processing stages that focus on H. pylori activity regions, which are lumens and the neighbouring regions surrounding tissue.  Training is performed with the annotation of a subset of the training images, which are used for training the model, which is then used to generate new candidates for annotation.  Using 9 WSI, which produce around 35K positive samples and 3.5M negative samples, the method shows a sensitivity 97.1% and a specificity of 99.9%.

The paper shows an interesting application of deep learning models, but there are a few issues.  First, there is essentially no reference to previous work developed in the field, and it is worth noting that one of the first deep learning papers in medical image analysis that has become well known in the field focuses on a similar problem - here is the paper:
Cireşan DC, Giusti A, Gambardella LM, Schmidhuber J. Mitosis detection in breast cancer histology images with deep neural networks. InInternational Conference on Medical Image Computing and Computer-assisted Intervention 2013 Sep 22 (pp. 411-418). Springer, Berlin, Heidelberg.  There have been many more papers after that, but this submission fails to acknowledge that.  Second, the experimental setup is unclear, particularly with respect to the training images used and the split between training and testing.  Also, how was the selection process used to define the testing samples?  In addition, how relevant are these results for a clinical application?  Finally, the pre-processing stages that focus on H. pylori activity regions is quite unclear.

Minor issues:
- Please clarify the term “deep learning of the artificial intelligence” in the title.
- Can you provide appropriate references for this sentence in the introduction: “In the
field of image recognition, the Convolutional Neural Network (CNN) is very prominent and can
even surpass the normal human performance in many specific scenarios”?
- Please fix the typos in the last two sentences of Section 2.1 (sentences finishing with “:”).  In fact the last sentence in Section 2.1 is unclear and needs to be revised.
- In section 2.1, the term “vague” is unclear in the sentence: “A large number of H. pylori are too vague to be visually recognized”.
- Please indicate in the images of Figure 3 where the H.pylori is located.
- Is there any overlap between the training and testing sets of WSIs?
- Please clarify what each image in Figures 4,5,6 represents.
- The method in Section 2.5.1 could be summarised using a proper algorithm representation in order to improve its reproducibility.
- Please provide a more convincing explanation behind the use of a threshold 0.7.  How was it defined exactly?  Was it defined with the training set?  Was it defined based on some sensitivity / specificity criterion?

**Special Issue:**

No

---

### Review · AnonReviewer3 · 2018-05-09
**Out of date methods, insufficient experiments and flawed writing**

**Rating:** 1
**Confidence:** 3

**Review:**

This paper presents a pipeline using a small 2D CNN to detect Helicobacter pyloric organism from whole slide images (WSIs). The detection problem is formulated into a patch-classification problem.

The paper was written with flawed scientific writing and thus hard to read. There are too many typos and grammatic errors, even in the paper title (Helicobector -> Helicobacter) and the abstract that make the paper insufficient for publishing.

The methods proposed in this paper is out-dated and not novel for detection problems. Hardly any recent related work in neither the object detection nor the pathology image analysis were mentioned or compared.

The experiments reported only the confusion matrix regarding the extracted patches. However, for such detection problems, it makes more sense to report the FROC curves. There is no comparison with the deep learning solutions from the recent studies on similar topics. It is thus impossible to measure the value of the reported numbers on an in-house dataset.

**Special Issue:**

No

---

### Review · AnonReviewer1 · 2018-05-09
**This paper presents the detection method of Helicobacter pylori (H. pylori) on whole slide images from gastric biopsy. Although they have defined an interesting problem, there is a lack of experiments results for evaluating algorithm and there is no technical novelty.**

**Rating:** 2
**Confidence:** 3

**Review:**


Quality & Clarity

#1. This paper is well organized for description of proposed system except for some typos in sub-titles.
  - “2.3 Data augment” -> Data augmentation
  - What is meaning of “2.5 Machine detection”.
#2. There is a lack of experimental results. The results of proposed system should be shown step by step.
  - The sensitivity of patch extraction needed.
  - The test set is very small dataset.
#3. The description of some figures and equation should be complemented.
  - In Figure 3, what is the “manual selection” for sampling of negative patches?
  - The description of Fig. 7 is ambiguous.
  - The loss function represented in Equation 1 is a general function.

Originality & Significance

(+) Multi-layer scanning has been performed for acquisition of H.pylori.
(+) Author proposed efficient annotation system based on initial trained network because pathologists aren’t able to annotate all regions of whole slide images.
(-) There is a weak technical novelty. In the CNN training step, they used traditional methods such as CIFAR10 network.
(-) In order to verify the originality of the proposed system, they need to experiment on the performance improvement according to the method. They only reported the final performance of the proposed system.

**Special Issue:**

No

---

### Decision · Program_Chairs · 2018-05-15
**Paper18 Acceptance Decision**

Reject